# ADVERSARIAL FEATURE MAP PRUNING FOR BACK-DOOR

**Dong Huang**[1]*, **Qingwen Bu**[2,3]*
[1]University of Hong Kong, [2]Shanghai AI Laboratory, [3]Shanghai Jiao Tong University
dhuang@cs.hku.hk, qwbu01@sjtu.edu.cn

## ABSTRACT

Deep neural networks have been widely used in many critical applications, such as autonomous vehicles and medical diagnosis. However, their security is threatened by backdoor attacks, which are achieved by adding artificial patterns to specific training data. Existing defense strategies primarily focus on using reverse engineering to reproduce the backdoor trigger generated by attackers and subsequently repair the DNN model by adding the trigger into inputs and fine-tuning the model with ground-truth labels. However, once the trigger generated by the attackers is complex and invisible, the defender cannot reproduce the trigger successfully then the DNN model will not be repaired, as the trigger is not effectively removed.

In this work, we propose Adversarial Feature Map Pruning for Backdoor (FMP) to mitigate backdoor from the DNN. Unlike existing defense strategies, which focus on reproducing backdoor triggers, FMP attempts to prune backdoor feature maps, which are trained to extract backdoor information from inputs. After pruning these backdoor feature maps, FMP will fine-tune the model with a secure subset of training data. Our experiments demonstrate that, compared to existing defense strategies, FMP can effectively reduce the Attack Success Rate (ASR) even against the most complex and invisible attack triggers (e.g., FMP decreases the ASR to 2.86% in CIFAR10, which is 19.2% to 65.41% lower than baselines). Second, unlike conventional defense methods that tend to exhibit low robust accuracy (that is, the accuracy of the model on poisoned data), FMP achieves a higher RA, indicating its superiority in maintaining model performance while mitigating the effects of backdoor attacks (e.g., FMP obtains 87.40% RA in CIFAR10). Our code is publicly available at: https://github.com/hku-systems/FMP.

## 1 INTRODUCTION

Deep neural networks (DNNs) have become a cornerstone technology in numerous fields, such as computer vision (Russakovsky et al., 2014), natural language processing (Devlin et al., 2019), speech recognition (Park et al., 2019), and several other applications (Bojarski et al., 2016; Rajpurkar et al., 2017). The effective training of DNNs generally demands extensive datasets and considerable GPU resources to achieve SOTA performance. However, a substantial number of DNN developers may lack access to these resources, which subsequently leads them to rely on third-party services for training their models (e.g., Google Cloud Google (2023), AWS Amazon Web Services (2023), and Huawei Cloud Huawei Technologies Co. (2023)), acquiring datasets (e.g., DataTurks DataTurks (2023)), or directly downloading pre-trained models (e.g., Hugging Face Face (2023)).

Although using third-party services offers a practical solution for DNN developers, it simultaneously introduces potential security risks. Specifically, third-party may be involved in the data collection, model training, and pre-trained model distribution process, which may introduce malicious backdoor triggers (Chen et al., 2017; Gu et al., 2017). For instance, once a developer uploads their dataset to a third-party service for training, the service provider could potentially revise the dataset by injecting poisoned samples containing hidden backdoor triggers. These poisoned samples, designed to blend in with the rest of the dataset, are then used in the training process, embedding the backdoor triggers into the model's architecture. The presence of backdoor triggers in DNNs can have serious

---

*Equal contribution.

implications, particularly in security-critical systems where model integrity is crucial to preserving human life and safety (Gu et al., 2017; Wang et al., 2019a).

To mitigate the backdoor threat, researchers have proposed a variety of defense mechanisms. Existing defense methods against backdoor attacks can be grouped into two main categories: detection methods and repair methods (Wu et al., 2022; Li et al., 2020a). Detection methods rely on internal information from the model (e.g. neuron activation values Chen et al. (2018)) or model properties (e.g., performance metrics (Chen et al., 2022; Zheng et al., 2022a)) to determine whether a model has been compromised by a backdoor attack, or whether a specific input example being processed by the model is a backdoor instance (Zheng et al., 2022b). Detection techniques are essential for pinpointing risks and highlighting security vulnerabilities. However, upon identifying a backdoor model, it must be promptly eliminated to regain trustworthiness and reliability.

To tackle this challenge, researchers have introduced repairing methods that go beyond detection and aim to remove backdoor triggers from the compromised models. One notable example is Neural Cleanse (NC) (Wang et al., 2019a), a technique that leverages reverse engineering to first reproduce the backdoor trigger. Once the trigger has been identified, NC injects it into the dataset along with its corresponding ground-truth labels, enabling the fine-tuning of the model to eliminate the backdoor trigger's impact. However, recent evaluation results (Chen et al., 2017; Li et al., 2020b; Nguyen & Tran, 2021) reveal that NC and similar reverse engineering-based methods are predominantly successful in tackling simple backdoor triggers. In contrast, complex triggers (e.g. those involving intricate patterns or transformations) pose a greater challenge, making it difficult to reproduce and remove them. Consequently, models containing such sophisticated triggers may not be adequately repaired, leaving them susceptible to potential security breaches.

Recently, some researchers (Liu et al., 2018; Zhao & Lao, 2022) try to address this problem by analyzing the feature map's behavior to remove the backdoor from the model since they notice that backdoor-related feature maps exist, which demonstrate different characteristics from other normal feature maps. Specifically, feature maps are essential components of DNNs, responsible for extracting various features (e.g., color and texture information) from input samples (Zeiler & Fergus, 2013). The model then relies on these feature maps to extract features from the inputs and to make its final predictions. In the backdoor model, some backdoor feature maps extract backdoor information from the inputs (Zhao & Lao, 2022). Based on this observation, they believe that once the backdoor feature maps in the model are erased, the backdoor trigger will also be removed from the model. However, detecting backdoor feature maps from the model is challenging since the DNN developers, who may download pre-trained models from a third party, are unaware of the information about possible backdoor triggers injected in the model. Subsequently, they can not directly add triggers into the input samples to analyze and detect backdoor feature maps.

To address this challenge, we propose the Adversarial Feature Map Pruning for Backdoor (FMP), a novel approach that focuses on identifying and eliminating backdoor feature maps within the DNN. Instead of directly reproducing the backdoor trigger and then using it to detect backdoor feature maps in the model, FMP uses adversarial attack to reproduce the features extracted by each feature map. By adding these features into the inputs, we can feed these inputs into the model to identify the backdoor feature maps, which can then be mitigated to secure the DNN model. Our experiments reveal that initializing these backdoor-propagating feature maps and fine-tuning the model can effectively neutralize the backdoor from the model.

To validate the effectiveness of the FMP, we conduct extensive experimental evaluations on multiple benchmark datasets, including CIFAR-10, CIFAR-100 (Krizhevsky & Hinton, 2009), and GT-SRB (Stallkamp et al., 2012), under diverse attack scenarios. In our experiments, we consider various backdoor triggers with different complexities and compare the performance of FMP against several state-of-the-art backdoor defense methods. Models are evaluated with three primary metrics: Accuracy, Attack Success Rate (ASR), and Robust Accuracy (RA).

Our experiment results demonstrate that FMP consistently achieves lower ASR, higher accuracy, and improved RA compared to baseline methods across different datasets and attack scenarios. For instance, on the CIFAR-10 dataset, FMP outperformed baselines by achieving a significantly lower ASR (e.g., FMP decrease 19.2% ASR average compared with baselines in CIFAR10), and improved RA (e.g., FMP increase 16.92% RA on average in CIFAR10), indicating its effectiveness in removing backdoor triggers without compromising the model's performance on benign samples.

In a nutshell, our contribution can be summarized as follows:

- We propose a novel defense strategy FMP to mitigate backdoor triggers from the model.

- We conduct substantial evaluations to investigate the performance of FMP. The experiment results show that FMP can significantly outperform previous defense strategies.

## 2 RELATED WORK

### 2.1 BACKDOOR ATTACKS

According to the threat model, existing backdoor attack methods can be partitioned into two general categories, including data poisoning and training controllable.

Data poisoning attacks entail manipulating training data to compromise machine learning models. Techniques in this realm aim to enhance the stealthiness and efficacy of attacks by designing triggers with specific characteristics. These triggers vary in visibility, locality, additivity, and sample specificity. Visible triggers, such as those in BadNets (Gu et al., 2017), visibly alter inputs, while invisible ones, like Blended (Chen et al., 2017), remain covert. Local triggers, as seen in Label Consistent Attack (Turner et al., 2019), affect small input regions, whereas global triggers, e.g., in SIG (Barni et al., 2019), impact broader areas. Additive triggers, such as Blended (Chen et al., 2017), add patterns to inputs, while non-additive ones, like LowFreq (Zeng et al., 2021), involve more intricate manipulations. In contrast to sample-agnostic triggers that which remain constant across all poisoned samples, SSBA (Li et al., 2020b) customizes triggers for each specific input.

Training controllable attacks (*e.g.*, WaNet (Nguyen & Tran, 2021)) involve an attacker having simultaneous control over the training process and data, enabling them to jointly learn the trigger and model weights. This allows for potentially more effective and stealthy backdoor attacks.

The described backdoor attack presents a major threat to the reliability of DNN models, compromising their accuracy and leading to harmful misclassifications. To tackle this issue, we introduce FMP, a new defense method aimed at detecting and neutralizing the impact of backdoor attacks by identifying and eliminating backdoor feature maps within the DNN model.

### 2.2 PRIOR WORK ON BACKDOOR DEFENSES.

The research community has proposed a variety of defense mechanisms to detect and mitigate backdoor attacks in deep neural networks. These defenses can be broadly categorized into two groups: data-centric and model-centric approaches.

Data-centric defenses primarily focus on detecting and cleansing poisoned data points in the training dataset to prevent the model from learning the backdoor trigger. One such technique is the AC proposed by Chen et al. (2018), which identifies and removes poisoned data points by clustering activations of the penultimate layer. However, once the DNN developers rely on the third party to train the DNN model, they can not use these strategies to remove backdoor samples since they can not interfere with the third-party platforms.

Model-centric defenses aim to address backdoor attacks by directly analyzing and modifying the trained model itself. NC (Wang et al., 2019a) is an example of such defense, which identifies and mitigates backdoor triggers by assessing anomaly scores based on adversarial perturbations. However, its effectiveness diminishes when dealing with complex and imperceptible triggers, limiting its utility for developers in reproducing and removing such triggers successfully. The Fine-Pruning (Liu et al., 2018) approach, as mentioned earlier, removes backdoors by pruning redundant feature maps less useful for normal classification. NAD (Li et al., 2021a) identifies backdoor neurons by analyzing the attribution of each neuron's output with respect to the input features. Nonetheless, the absence of backdoor or poison samples from injected datasets can impede accurate analysis, reducing their efficacy in trigger removal. Several recent studies (Liu et al., 2019; Barni et al., 2019; Chen et al., 2019; Guo et al., 2019; Fu et al., 2020; Wu & Wang, 2021; Guo et al., 2021; Bu et al., 2023; Li et al., 2023; Liang et al., 2023) are also proposed to defense backdoor for deep neural networks.

## 3 METHODOLOGY

**Notations** For ease of discussion, this section defines the following notations for DNNs and feature maps: $f_\theta$ is a DNN parameterized by $\theta$, and there are $N$ feature maps $\sum_i^N f_\theta^i$ in the model. The $i$-th feature map in the DNN is denoted as $f_\theta^i$. $f_\theta^i(x)$ denotes the feature map $i$-th output when the DNN input is $x$. The $x'$ means the input x added the feature generated by the corresponding feature map.

### 3.1 MOTIVATION

The primary goal of our pruning defense strategy is to detect backdoor feature maps in a DNN model. As previously mentioned, intuitively, we should use backdoor samples and clean samples fed into the model to detect these backdoor feature maps. However, since DNN developers do not have access to backdoor samples, they cannot directly detect backdoor feature maps using this approach. This limitation calls for a defense mechanism that can operate effectively without the need for backdoor samples.

The main idea behind our proposed defense mechanism is to generate potential poison samples by reversing the specific features associated with each feature map. Since each feature map in the model is used to extract features from the DNN model, and in the injected model, there are some backdoor feature maps that are used to extract backdoor information for the poison samples. By feeding these potential poison samples into the model, we can observe the inference accuracy for each feature map. Since backdoor feature maps are designed to extract backdoor information from the inputs, causing the classifier to return the target label regardless of the true label, we hypothesize that potential poison samples generated by backdoor feature maps will lead to a significant change in inference accuracy when processing their corresponding potential poison samples.

In summary, the motivation behind our defense strategy is to detect backdoor feature maps without backdoor samples by generating potential poison samples and analyzing the inference accuracies with these samples. This enables us to identify backdoor feature maps, which can be mitigated to secure the DNN model. With this motivation in mind, we propose Adversarial Feature Map Pruning for Backdoor (FMP) as the defense mechanism to achieve this objective.

### 3.2 FEATURE REVERSE GENERATION

In this section, we employ reverse engineering to generate features that will be extracted by each feature map in the DNN. These features will be added to the inputs, which will then be used to detect backdoor feature maps. The detailed implementation of the Feature Reverse Generation is provided in Algorithm 1. Specifically, for each feature map $f_\theta^i$ in the DNN model, our objective is to generate the corresponding features that the feature map is intended to extract by adversarial feature map attack. To accomplish this, we use a reverse engineering approach that maximizes the difference between the original input $x$ and the perturbed input $x'$ in the feature map space. In Algorithm 1, referred to as $FRG$, we have adapted the FGSM attack (Goodfellow et al., 2014) to operate at the feature map levels. The overall time required by this method can be expressed as follows:

$$T = L \cdot forward + N_i \cdot backward$$

where $L$ represents the total number of layers in the Deep Neural Network (DNN), and $N_i$ denotes the number of feature maps in the $i$-th layer. The term "forward" corresponds to the time taken for one forward step, assuming each forward pass takes an equal amount of time. Similarly, the term "backward" corresponds to the time taken for one backward step, assuming each backward pass also takes an equal amount of time.

### 3.3 REMOVE BACKDOOR FROM THE DNN

After obtaining the reversed samples using the method described in Algorithm 1, we feed these potential poison samples into the model to obtain feature map inference accuracy. Then, based on the accuracy list, we will return the potential backdoor injected feature maps. After detecting the backdoor-related feature maps, our next step is to remove the backdoor from the DNN. To achieve this, we follow a two-step process: initialization and fine-tuning.

---

**Algorithm 1:** Feature Reverse Generation

---

**input** : $f_\theta$: DNN model; $(\mathcal{X}, \mathcal{Y})$: Clean Dataset; $\epsilon$: maximum allowed perturbation; $p$: First
$N/p$ feature maps wll be pruned.

**output:** List: Backdoor feature map list.

1 **Function** *Inference*$(f_\theta, (\mathcal{X}, \mathcal{Y}), \epsilon, p)$**:**
2     Initialize an empty list *Correct_list*;
3     Start traversal over all feature maps: ;
4     **for** *i in range(N)* **do**
5        Initialize *correct* counter to 0;
6        **for** *x, y in* $(\mathcal{X}, \mathcal{Y})$ **do**
7           $x' = \text{FRG}(f_\theta^i, x, steps, \alpha, \epsilon)$;
8           $y' = f_\theta(x')$;
9           **if** $y = y'$ **then**
10              *correct* += 1;
11        Append *correct* to *Correct_list*;
12     Sort *Correct_list* in ascending order;
13     **return** First $N/p$ elements from *Correct_list* ;
14     *# We suppose feature maps with lower FRG accuracy are backdoor-related feature maps.*
15 **Function** *FRG*$(f_\theta^i, x, \epsilon)$**:**
16     Initialize: $x' = x + $ random noise ;
17     Calculate the loss: $loss = \|f_\theta^i(x) - f_\theta^i(x')\|^2$;
18     Calculate the gradient: $grad = \partial loss / \partial x$;
19     Update: $x' = x + \epsilon \cdot \text{sign}(grad)$;
20     Apply image bounds: $x' = \text{clamp}(x', \min = 0, \max = 1)$;
21     **return** $x'$;

---

**Initialization.** In this phase, we start by setting the weights of the identified backdoor feature maps to zero, effectively neutralizing their influence on the model's output. By doing this, we eliminate the backdoor information these maps were designed to capture. Additionally, we adjust the biases of these feature maps to keep their output close to zero during forward pass. This initialization process helps mitigate the impact of malicious backdoors on the model's predictions, lowering the risk of targeted misclassification.

**Fine-Tuning.** After initializing the backdoor feature maps, the entire DNN model undergoes fine-tuning. This process adjusts the model weights with a reduced learning rate and a subset of accurately labeled samples. Fine-tuning enables the model to adapt to the new initialization while maintaining its performance on clean samples. This ensures the model's predictive accuracy on clean inputs and bolsters its resilience against potential attacks.

## 4 EVALUATION

**Experimental Setting** In this work, we use BackdoorBench (Wu et al., 2022) as a benchmark to evaluate the performance of FMP. Since most of the existing backdoor-related literature focused on image classification tasks (Chen et al., 2017; 2022; Devlin et al., 2019; Zheng et al., 2022b; Zhao & Lao, 2022; Wang et al., 2019b; Tran et al., 2018; Nguyen & Tran, 2021), we followed existing research to the choice of CIFAR10, CIFAR100 (Krizhevsky & Hinton, 2009), and GTSRB (Stallkamp et al., 2012) datasets to evaluate FMP's performance. Similarly to baselines, we chose ResNet18 (He et al., 2015) as our evaluation model, as it has been systematically evaluated by BackdoorBench (Wu et al., 2022), widely used by baseline approaches, and is also widely used by vision tasks (e.g., image classification (He et al., 2015), object detection (Ren et al., 2015)), so we believe evaluating FMP in these data sets can provide a comprehensive and meaningful assessment of its performance. In our experiments, we selected five state-of-the-art backdoor attack strategies, which are systematically evaluated and implemented by BackdoorBench, as baselines to evaluate the effectiveness of our defense strategy. These attack strategies consist of BadNets (Gu et al., 2017), Blended (Chen et al.,

2017), Low Frequency (Zeng et al., 2021), SSBA (Li et al., 2020b), and WaNet (Nguyen & Tran, 2021). We train the CIFAR10 and CIFAR100 datasets with 100 epochs, SGD momentum of 0.9, learning rate of 0.01, and batch size of 128, using the CosineAnnealingLR scheduler. The GTSRB dataset is trained with 50 epochs. We set the poisoning rate to 10% by default. To ensure fair comparisons, we adhere to the default configuration in the original papers, including trigger patterns, trigger sizes, and target labels.

**Defense setup** BackdoorBench has implemented many effective defense strategies (Zhao & Lao, 2022; Liu et al., 2018; Wang et al., 2019a; Li et al., 2021a; Tran et al., 2018) are implemented by BackdoorBench, we believe that taking these defense strategies as our baseline can demonstrate FMP's effectiveness. However, some of these defense strategies require DNN developers to have access to the backdoor trigger (e.g., AC (Chen et al., 2018), Spectral (Tran et al., 2018), ABL (Li et al., 2021b), D-BR (Chen et al., 2022) and DDE (Zheng et al., 2022b)), or to modify the model training process (e.g., ABL (Li et al., 2021b), DBD (Huang et al., 2022)) which is not possible in our defense scenarios. Therefore, we exclude these strategies from our baseline comparison.

Finally, we select six widely used defense strategies that align with our defense goals: *Fine-tuning (FT)* retrains the backdoor model with a subset clean dataset to remove the backdoor trigger's effects. *Fine-pruning (FP)* (Liu et al., 2018) prunes backdoor feature maps to remove backdoors from the model. *Adversarial Neuron Pruning (ANP)* (Wu & Wang, 2021) selectively prunes neurons associated with the backdoor trigger while maintaining performance. *Channel Lipschitz Pruning (CLP)* (Zhao & Lao, 2022) is a data-free strategy that analyzes the Lipschitz constants of the feature maps to identify potential backdoor-related feature maps. *Neural Attention Distillation (NAD)* (Li et al., 2021a) leverages attention transfer to erase backdoor triggers from deep neural networks, ensuring that the model focuses on the relevant neurons for classification. *Neural Cleanse (NC)* identifies and mitigates backdoor attacks in neural networks by analyzing the internal structure of the model and identifying patterns that are significantly different from the norm, which may indicate the presence of a backdoor.

In order to repair the model, we adopt a configuration consisting of 10 epochs, a batch size of 256, and a learning rate of 0.01. The CosineAnnealingLR scheduler is employed alongside the Stochastic Gradient Descent (SGD) optimizer with a momentum of 0.9 for the client optimizer. We set the default ratio of the retraining data set at 10% to ensure a fair and consistent evaluation of defense strategies. For the CLP, we configure the Lipschitz Constant threshold ($u$) to be 3, the pruning step to 0.05, and the maximum pruning rate to 0.9, which is consistent with BackdoorBench and its original paper default settings. In the case of ANP, we optimize all masks using Stochastic Gradient Descent (SGD) with a perturbation budget (i.e., $\epsilon$) of 0.4. For FMP, we set $p$ equal to 64 and $\epsilon$ as 1/255. All other configurations are maintained as specified in the respective original publications to guarantee a rigorous comparison of the defense strategies.

**Evaluation Metrics** To evaluate our proposed defense strategy and compare it with SoTA methods, we utilize three primary evaluation metrics: clean accuracy (Acc), attack success rate (ASR), and robust accuracy (RA). Acc gauges the defense strategy's performance on clean inputs, reflecting whether the model's original task performance remains intact. ASR assesses the defense strategy's effectiveness against backdoor attacks by measuring the rate at which backdoor triggers succeed in causing misclassification. RA is a complementary evaluation metric that focuses on the model's performance in correctly classifying poisoned samples despite the presence of backdoor triggers.

### 4.1 EFFECTIVENESS

The evaluation results are shown in Tab. 1. In analyzing experiments with the CIFAR10 dataset, it is evident that FMP consistently maintains a low Attack Success Rate (ASR) alongside a high Robust Accuracy (RA). Specifically, FMP achieves an average ASR of 2.86%, significantly reducing the initial 19.2% ASR in CIFAR10. Additionally, the RA of FMP surpasses that of baseline strategies, reaching 87.40%, which is approximately 16.92% higher than the average RA of baseline approaches. This indicates the effectiveness of our approach in mitigating backdoor attacks and enhancing model performance on poisoned data, demonstrating applicability in real-world scenarios.

We can also observe that standard fine-tuning (FT) shows promising defense results against several attacks, such as BadNets, where it achieves an ASR of 1.51% and an RA of 92.46%. However, it fails

Table 1: **Performance comparison (%) of backdoor defense methods on CIFAR10, CIFAR100, and GTSRB datasets with PreActResNet18 model.** We perform evaluations under five advanced attacks with a poison rate of 10% and the retraining data ratio set to 100%. The $\epsilon$ and $p$ are set to 1/255 and 64 respectively (Algorithm 1). FMP achieves leading performance across various settings.

| Methods | BadNets Acc | ASR | RA | Blended Acc | ASR | RA | Low Frequency Acc | ASR | RA | SSBA Acc | ASR | RA | WaNet Acc | ASR | RA | AVG Acc↑ | ASR↓ | RA↑ |
|---|---|---|---|---|---|---|---|---|---|---|---|---|---|---|---|---|---|---|
| Benign | 91.94 | 97.21 | - | 93.44 | 99.95 | - | 93.44 | 99.39 | - | 92.94 | 98.80 | - | 91.53 | 98.83 | - | 92.65 | 98.84 | - |
| FT | 93.35 | **1.51** | **92.46** | 93.54 | 95.63 | 4.16 | 93.41 | 77.89 | 20.67 | 93.28 | 71.57 | 26.52 | 94.12 | 22.30 | 74.53 | **93.54** | 53.78 | 43.67 |
| FP | 91.62 | 16.06 | 79.24 | 93.21 | 96.84 | 3.04 | 93.24 | 96.73 | 2.79 | 92.88 | 82.92 | 16.00 | 91.30 | **0.62** | 84.91 | 92.45 | 58.63 | 37.20 |
| CLP | 91.20 | 94.77 | 5.01 | 93.29 | 99.83 | 0.17 | 92.47 | 99.10 | 0.82 | 88.27 | 9.96 | 76.22 | 89.03 | 53.84 | 41.88 | 90.85 | 71.5 | 24.82 |
| ANP | 91.22 | 73.36 | 26.16 | 93.25 | 99.44 | 0.56 | 93.19 | 98.03 | 1.88 | 92.92 | 68.59 | 29.13 | 90.81 | 1.93 | 88.98 | 92.28 | 68.27 | 29.34 |
| NC | 89.11 | 1.32 | 89.17 | 93.23 | 99.90 | 0.10 | 90.67 | 2.26 | 86.24 | 90.33 | **0.53** | 86.72 | 90.54 | 86.91 | 12.38 | 90.78 | 38.18 | 54.92 |
| NAD | 92.97 | 1.10 | 92.39 | 92.18 | 44.98 | 42.60 | 92.32 | 13.43 | 77.03 | 92.22 | 38.18 | 57.18 | 94.16 | 12.61 | 83.22 | 92.77 | 22.06 | 70.48 |
| Our | 91.67 | 1.67 | 91.71 | 91.85 | **6.44** | **74.43** | 91.77 | **1.90** | **90.52** | 91.92 | 2.89 | **88.59** | 93.42 | 1.38 | **92.16** | 92.13 | **2.86** | **87.40** |

(a) CIFAR-10

| Methods | BadNets Acc | ASR | RA | Blended Acc | ASR | RA | Low Frequency Acc | ASR | RA | SSBA Acc | ASR | RA | WaNet Acc | ASR | RA | AVG Acc↑ | ASR↓ | RA↑ |
|---|---|---|---|---|---|---|---|---|---|---|---|---|---|---|---|---|---|---|
| Benign | 67.33 | 93.61 | - | 70.19 | 99.84 | - | 69.56 | 97.47 | - | 69.24 | 98.12 | - | 65.67 | 97.50 | - | 68.39 | 97.31 | - |
| FT | 69.90 | 1.38 | 67.18 | 70.32 | 89.36 | 6.57 | 69.93 | 45.90 | 35.95 | 69.30 | 58.14 | 28.45 | 71.16 | 6.41 | 63.64 | **70.12** | 40.23 | 40.35 |
| FP | 67.59 | 23.09 | 54.04 | 69.44 | 93.56 | 4.03 | 68.73 | 82.60 | 10.84 | 68.25 | 76.46 | 14.52 | 66.33 | 81.47 | 11.26 | 68.06 | 71.44 | 18.94 |
| CLP | 59.74 | 74.79 | 19.42 | 68.56 | 99.52 | 0.37 | 65.46 | 92.88 | 5.18 | 68.40 | 89.57 | 7.44 | 60.06 | 6.93 | 54.53 | 64.44 | 72.74 | 17.38 |
| ANP | 66.95 | 15.36 | 58.05 | 70.05 | 97.70 | 1.70 | 69.47 | 55.00 | 32.00 | 68.92 | 89.87 | 7.54 | 64.23 | 0.23 | 60.55 | 67.92 | 51.63 | 31.97 |
| NC | 64.67 | 0.10 | 64.38 | 64.16 | 1.14 | 34.63 | 65.25 | 2.28 | 53.91 | 65.16 | 2.17 | 56.86 | 67.09 | 1.29 | 62.92 | 65.27 | 1.39 | 54.54 |
| NAD | 68.84 | 0.31 | **67.63** | 69.24 | 81.07 | 10.16 | 69.33 | 31.78 | 44.41 | 68.39 | 30.82 | 42.24 | 71.57 | 17.41 | 57.72 | 69.47 | 32.27 | 44.43 |
| Our | 66.25 | **0.09** | 67.23 | 67.27 | **0.41** | 40.49 | 66.32 | **0.18** | 65.83 | 66.77 | **0.39** | 60.61 | 68.79 | **0.18** | 66.84 | 67.08 | **0.25** | 60.20 |

(b) CIFAR-100

| Methods | BadNets Acc | ASR | RA | Blended Acc | ASR | RA | Low Frequency Acc | ASR | RA | SSBA Acc | ASR | RA | WaNet Acc | ASR | RA | AVG Acc↑ | ASR↓ | RA↑ |
|---|---|---|---|---|---|---|---|---|---|---|---|---|---|---|---|---|---|---|
| Benign | 98.04 | 96.38 | - | 98.19 | 100.00 | - | 98.21 | 99.96 | - | 97.73 | 99.72 | - | 98.66 | 97.59 | - | 98.17 | 98.72 | - |
| FT | 98.60 | 0.49 | 98.11 | 98.19 | 99.61 | 0.34 | 98.39 | 92.94 | 5.43 | 97.63 | 96.75 | 2.99 | 99.20 | 0.73 | 98.31 | 98.40 | 58.10 | 41.03 |
| FP | 97.79 | 0.04 | 65.20 | 98.04 | 100.00 | 0.00 | 97.81 | 99.25 | 0.45 | 96.90 | 99.64 | 0.34 | 98.80 | 0.05 | 13.23 | 97.87 | 59.79 | 15.84 |
| CLP | 96.42 | 88.11 | 11.67 | 97.68 | 100.00 | 0.00 | 97.09 | 97.74 | 1.82 | 97.13 | 99.42 | 0.57 | 62.17 | 99.91 | 0.08 | 90.10 | 97.03 | 2.83 |
| ANP | 96.55 | 15.71 | 82.06 | 98.22 | 99.96 | 0.02 | 98.27 | 69.20 | 27.66 | 97.10 | 99.51 | 0.48 | 98.67 | 1.65 | 96.32 | 97.76 | 57.20 | 41.30 |
| NC | 97.75 | 0.00 | 97.75 | 96.34 | 2.47 | 53.58 | 97.72 | 7.29 | 81.19 | 96.94 | 3.70 | 88.40 | 98.39 | 0.00 | 97.29 | 97.43 | 2.69 | 83.64 |
| NAD | 98.66 | 0.03 | 98.64 | 98.39 | 96.98 | 2.86 | 98.54 | 80.88 | 14.91 | 97.72 | 94.70 | 4.93 | 98.98 | 0.16 | 98.66 | **98.45** | 54.55 | 44.00 |
| Our | 98.60 | **0.00** | **98.66** | 90.36 | **1.07** | **64.17** | 90.01 | **0.02** | **95.58** | 97.41 | **0.51** | **89.87** | 99.05 | **0.00** | **98.93** | 95.08 | **0.32** | **89.44** |

(c) GTSRB

to generalize to more complex attacks, like the Blended attack, where the ASR increases to 95.63%, and the RA drops to 4.16%. Neural Attention Distillation (NAD) exhibits similar behavior in terms of generalization, with its performance varying considerably across different attacks. For example, NAD achieves an ASR of 1.10% and an RA of 92.39% against BadNets, but its performance drops significantly when faced with the Blended attack, resulting in an ASR of 44.98%.

For pruning-based methods, for example, FP, Adversarial Neuron Pruning (ANP), and Channel Lipschitz Pruning (CLP), these approaches demonstrate varying levels of effectiveness against different attacks. However, none of them consistently outperform our method FMP in all types of attacks. For example, FP achieves an ASR of 16.06% and an RA of 79.24% against BadNets, while ANP achieves an ASR of 73.36% and an RA of 26.16% for the same attack when it only has a limited fine-tune dataset (10%) and within limited fune-tune epochs (i.e., 10 epochs). The primary factor contributing to the superior performance of FMP over other pruning methods lies in its alignment with the backdoor trigger's focus on the DNN feature map rather than specific neurons within the DNN, such as ANP. This alignment occurs at the feature map level, distinguishing FMP from other techniques and facilitating its efficacy in achieving higher performance levels.

For the reverse engineering-based method (i.e., NC), we can observe that when the backdoor triggers are simple and easy to reproduce, NC will have high performance, e.g., NC achieves an ASR of

Table 2: **FMP's effectiveness under different poison rates.** Our approach exhibits consistent performance across different poison ratios, highlighting its generalizability and reliability.

| Poison Rate (%) | BadNets | | | Blended | | | Low Frequency | | | SSBA | | | WaNet | | |
|---|---|---|---|---|---|---|---|---|---|---|---|---|---|---|---|
| | Acc | ASR | RA | Acc | ASR | RA | Acc | ASR | RA | Acc | ASR | RA | Acc | ASR | RA |
| 0.1 | 92.09 | 1.71 | 91.72 | 91.83 | 8.53 | 75.90 | 91.74 | 2.09 | 91.00 | 92.17 | 2.23 | 89.60 | 93.19 | 1.48 | 91.91 |
| 0.5 | 92.01 | 0.99 | 91.74 | 91.95 | 8.31 | 75.30 | 91.92 | 2.14 | 91.21 | 91.90 | 1.42 | 91.92 | 93.32 | 1.45 | 91.94 |
| 1 | 92.22 | 1.78 | 91.48 | 91.66 | 8.49 | 74.49 | 92.06 | 2.08 | 91.00 | 91.85 | 2.08 | 89.61 | 93.46 | 1.32 | 92.22 |
| 5 | 92.59 | 1.24 | 90.39 | 93.45 | 8.31 | 74.47 | 93.32 | 2.23 | 90.07 | 93.10 | 2.93 | 88.49 | 91.86 | 1.09 | 91.90 |
| 10 | 91.67 | 1.67 | 91.71 | 91.85 | 6.44 | 74.43 | 91.77 | 1.90 | 90.52 | 91.92 | 2.89 | 88.59 | 93.42 | 1.38 | 92.16 |

Table 3: **FMP's performance with different retraining data ratios.** Expanding the retraining dataset can consistently lower the attack success rate, yet our approach remains effective in neutralizing the backdoor impact even with just 5% of the data.

| Retraining ratio (%) | BadNets | | | Blended | | | Low Frequency | | | SSBA | | | WaNet | | |
|---|---|---|---|---|---|---|---|---|---|---|---|---|---|---|---|
| | Acc | ASR | RA | Acc | ASR | RA | Acc | ASR | RA | Acc | ASR | RA | Acc | ASR | RA |
| 5 | 86.57 | 1.77 | 86.08 | 90.59 | 7.63 | 63.40 | 88.07 | 4.60 | 81.00 | 87.26 | 3.50 | 79.77 | 89.31 | 1.54 | 88.07 |
| 10 | 91.67 | 1.67 | 91.71 | 91.85 | 6.44 | 74.43 | 91.77 | 1.90 | 90.52 | 91.92 | 2.89 | 88.59 | 93.42 | 1.38 | 92.16 |
| 15 | 91.71 | 1.66 | 91.30 | 91.88 | 6.37 | 74.61 | 91.60 | 1.88 | 90.63 | 91.95 | 2.87 | 88.64 | 92.47 | 1.42 | 91.32 |
| 20 | 91.83 | 1.47 | 91.91 | 91.92 | 6.08 | 74.87 | 91.74 | 1.90 | 90.73 | 91.73 | 2.88 | 88.91 | 92.54 | 1.44 | 91.33 |
| 100 | 92.02 | 0.9 | 91.04 | 92.12 | 5.31 | 75.77 | 91.24 | 1.71 | 91.07 | 92.31 | 2.67 | 89.37 | 92.95 | 1.37 | 91.42 |

1.32% and an RA of 89.17% against BadNets. However, when the backdoor triggers are complex and invisible, the performance of NC significantly deteriorates, indicating its limited effectiveness in handling sophisticated backdoor attacks. For instance, when faced with the Blended attack, where the backdoor trigger is more intricate and harder to detect, NC's ASR increases to an alarming 99.90%, and its RA plummets to a mere 0.10%.

For the CIFAR100 and GTSRB datasets, we can observe that first, FMP is also better than the baseline defense strategies. Specifically, FMP obtains the average 0.25% ASR and the average 60.2% RA in CIFAR100, which achieves an average ASR reduction of 1.14%. Furthermore, FMP also obtains a mean ASR of 0.32% and an average RA of 89.44% in GTSRB, showcasing a significant enhancement of approximately 5.80% in RA compared to baseline methods Furthermore, it is important to highlight the inconsistency in performance exhibited by baseline methods. For example, the NC technique displays an ASR of 1.14% and an RA of 34.63% when challenged by BadNets on CIFAR100. However, its effectiveness diminishes notably when subjected to the same attack but on a different dataset, such as CIFAR10. In contrast, the FMP approach consistently delivers superior results, consistently achieving lower ASR values across various datasets, including 2.86%, 0.25%, and 0.32% average ASR in CIFAR10, CIFAR100, and GTSRB, respectively.

**Effect of Poison Data Rate** The poison rate, referring to the proportion of poisoned samples in the training dataset, plays a crucial role in the results of the backdoor trigger injection. We conducted experiments with different poison rates (from 0.1% to 10%) to explore their impact on FMP's effectiveness. The results, shown in Tab.2, indicate that FMP demonstrates consistent performance across different poison rates and effectively mitigates backdoor attacks. For example, considering the BadNets attack, the ASR changes slightly within 0.99% to 1.78% as the poisoning rate increases from 0.1% to 10%. This trend is also observed for other attack strategies. Although a higher poison rate can be expected to lead to a higher ASR, our experimental results show that this is not true. When the poisoning rate is very low, it becomes more challenging for defense strategies to detect the backdoor trigger from the model due to its subtle influence. As the poison rate increases, the backdoor trigger has a more noticeable impact on the model, which can be detected and mitigated more easily by the defense strategy. Our experimental results emphasize the importance of designing defense strategies capable of detecting and mitigating backdoor attacks, even when dealing with subtle influences caused by low poison rates.

**Effectiveness under Different Percentages of Clean Data** We are also interested in studying the correlation between the performance of FMP and the amount of available training data, which will

Table 4: **FMP's effectiveness using different $\epsilon$ in CIFAR10 dataset under BadNets attack.** The value $\epsilon$ denotes the size of each optimization step for the perturbed samples in the FRG (Algorithm 1). FMP remains effective regardless of the specific parameter choice, yielding the best overall performance when $\epsilon$ is set to 1/255.

| $\epsilon$ | 1/255 | 2/255 | 4/255 | 8/255 | 16/255 |
|---|---|---|---|---|---|
| Acc | 91.67 | 91.04 | 90.28 | 89.21 | 88.01 |
| ASR | 1.67 | 1.22 | 1.51 | 2.29 | 1.41 |
| RA | 91.71 | 90.79 | 89.49 | 88.03 | 87.82 |

Table 5: **FMP's effectiveness using different $p$ in CIFAR10 dataset under BadNets attack.** Reducing the value of $p$ results in more extensive clipping of the feature map. For $p$ values greater than 4, FMP efficiently eliminates the backdoor and preserves the model's original accuracy.

| $p$ | 2 | 4 | 8 | 16 | 32 | 64 |
|---|---|---|---|---|---|---|
| Acc | 65.57 | 85.42 | 87.65 | 88.95 | 89.96 | 91.67 |
| ASR | 3.98 | 1.40 | 1.3 | 1.28 | 1.56 | 1.67 |
| RA | 65.52 | 85.49 | 87.63 | 88.78 | 90.34 | 91.71 |

be used to repair the model to mitigate backdoor triggers. We compare four different retraining data ratios:5%, 10%, 15%, 20%, and 100%, and the results of our FMP are demonstrated in Tab.3. We observe that the performance of our defense strategy improves as the amount of clean training data increases. For example, when the retraining ratio increases from 5% to 100%, the ASR for BadNets decreases from 1.77% to 0.9%, while the model accuracy (Acc) improves from 86.57% to 92.02% and the Robust Accuracy (RA) increases from 86.08% to 91.04%. Similar trends can be observed for other attack strategies such as Blended, Low Frequency, SSBA, and WaNet. This indicates that our defense strategy becomes more effective in mitigating backdoor attacks as more clean data are available to retrain the model. However, it should be noted that even with a small amount of clean data (e.g., 5%), our defense strategy still exhibits competitively good performance in mitigating backdoor attacks. For example, with a 5% retraining ratio, the ASR for WaNet is 1.54%, while the Acc and RA are 89.31% and 88.07%, respectively.

**Effectiveness under Different $\epsilon$ and $p$** We further investigate the effectiveness of FMP under different $\epsilon$ and $p$, as listed in Tab.4 and Tab.5. We can first observe that with different $\epsilon$, the effectiveness of FMP is consistently satisfactory. Upon increasing the $\epsilon$, the Acc exhibits marginal decline, underscoring FMP's resilience across varying $\epsilon$ values. Subsequently, when varying the parameter $p$ for backdoor feature pruning, a notable decrease is observed in both accuracy (Acc) and robust accuracy (RA). This reduction can be attributed to the model's failure to successfully finetune after 50% of the information is pruned along with 50% of the feature maps, hampering its performance. FMP can successfully execute to mitigate the backdoor from the model with a $p$ larger than 4.

## 5 CONCLUSION

In this paper, we presented a novel defense strategy, FMP, to effectively detect and mitigate backdoor attacks in DNNs. Our method targets the identification and removal of backdoor feature maps within the model. Through comprehensive experiments, we show the efficacy of our approach against various backdoor attack methods, surpassing current state-of-the-art defenses in terms of reducing attack success rates and maintaining stability across different datasets and attack methods.

## 6 ACKNOWLEDGEMENT

The work is supported in part by National Key R&D Program of China (2022ZD0160200), HK RIF (R7030-22), HK ITF (GHP/169/20SZ), the Huawei Flagship Research Grants in 2021 and 2023, and HK RGC GRF (Ref: HKU 17208223), the HKU-SCF FinTech AcademyR&D Funding Schemes

in 2021 and 2022, and the Shanghai Artificial Intelligence Laboratory (Heming Cui is a courtesy researcher in this lab).

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

# A    APPENDIX

## A.1    COMPARISON WITH OTHER BASELINES

To illustrate the effectiveness of FMP with other adversarial example related strategies (e.g., ANP Wu & Wang (2021), AEVA Guo et al. (2021), and RNP Li et al. (2023)). The evaluation results are shown in Tab. 6. We can find that FMP obtain SOTA performance compared with these baselines. For example, in BadNets, FMP decreases the ASR from 50.96% to 1.67%, and FMP also increases the RA from 47.53% to 91.71% in BadNets. The primary reason for this is FMP's focus on the feature map level, which aligns with the backdoor trigger's emphasis on the DNN feature map, rather than on specific neurons within the DNN (e.g., as in ANP). Secondly, the pruning at the feature map level enables FMP to rapidly eliminate the backdoor trigger from the model. In contrast, neuron-level pruning typically requires significant overhead due to the need for repeated prune-finetune-evaluation cycles. FMP is particularly advantageous in scenarios with limited computational resources, where developers may not have the capacity to extensively evaluate and remove backdoors from the model. This limitation results in strategies like ANP, AEVA, and RNP exhibiting higher ASR compared to FMP in our setup.

| Backdoor Attack | BadNets | | | Blended | | | Low Frequency | | | SSBA | | | WaNet | | |
|---|---|---|---|---|---|---|---|---|---|---|---|---|---|---|---|
| | Acc | ASR | RA | Acc | ASR | RA | Acc | ASR | RA | Acc | ASR | RA | Acc | ASR | RA |
| Benign | 91.94 | 97.21 | - | 93.44 | 99.95 | - | 93.44 | 99.39 | - | 92.94 | 98.80 | - | 91.53 | 98.83 | - |
| ANP | 91.22 | 73.36 | 26.16 | 93.25 | 99.44 | 0.56 | 93.19 | 98.03 | 1.88 | 92.92 | 68.59 | 29.13 | 90.81 | 1.93 | 88.98 |
| AEVA | 91.05 | 50.96 | 47.53 | 92.28 | 59.37 | 38.66 | 93.05 | 59.81 | 36.38 | 92.29 | 67.56 | 26.01 | 90.26 | 6.54 | 90.59 |
| RNP | 90.55 | 55.01 | 36.46 | 92.29 | 55.59 | 42.15 | 92.41 | 58.71 | 40.1 | 91.94 | 61.24 | 30.6 | 90.22 | 18.15 | 72.95 |
| FMP | 91.67 | **1.67** | **91.71** | 91.85 | **6.44** | **74.43** | 91.77 | **1.90** | **90.52** | 91.92 | **2.89** | **88.59** | 93.42 | **1.38** | **88.98** |

Table 6: Performance comparison (%) of adversarial example related backdoor defense methods on CIFAR10 under PreActResNet18, under different attack strategies with a poison rate of 10% and retraining data ratio of 100%. We set the $\epsilon$ to 1/255, and the $p$ is set to 64.

To illustrate the effectiveness of FMP with other SOTA backdoor defense strategies (e.g., DeepInspect Wu & Wang (2021), TABOR Guo et al. (2021), ABS Li et al. (2023) and Fu et al. (2020)). The evaluation results are shown in Tab. 7. We can find that FMP obtain SOTA performance compared with these baselines. For example, in BadNets, FMP decreases the ASR from 5.61% to 1.67%, and FMP also increases the RA from 79.02% to 91.71% in BadNets. As discussed before, the primary reason for this is the pruning at the feature map level enables FMP to efficiently and rapidly eliminate the backdoor trigger from the model. While other defense strategies will not obtain the SOTA performance due to the limited computational resources.

| Backdoor Attack | BadNets | | | Blended | | | Low Frequency | | | SSBA | | | WaNet | | |
|---|---|---|---|---|---|---|---|---|---|---|---|---|---|---|---|
| | Acc | ASR | RA | Acc | ASR | RA | Acc | ASR | RA | Acc | ASR | RA | Acc | ASR | RA |
| Benign | 91.94 | 97.21 | - | 93.44 | 99.95 | - | 93.44 | 99.39 | - | 92.94 | 98.80 | - | 91.53 | 98.83 | - |
| DeepInspect | 90.51 | 15.87 | 64.44 | 90.89 | 3.5 | 77.16 | 90.83 | 4.7 | 77.51 | 90.05 | 10.57 | 73.83 | 90.31 | 5.97 | 77.94 |
| TABOR | 90.78 | 9.19 | 79.02 | 90.78 | 11.13 | 78.22 | 90.14 | 5.48 | 76.44 | 90.03 | 13.09 | 76.71 | 90.95 | 7.36 | 78.29 |
| ABS | 90.78 | 5.61 | 77.34 | 90.95 | 14.44 | 80.33 | 90.93 | 10.43 | 88.32 | 90.95 | 17.12 | 77.45 | 90.6 | 13.04 | 77.6 |
| Fu et al. (2020) | 90.7 | 11.22 | 76.72 | 90.28 | 9.37 | 75.63 | 90.46 | 9.62 | 71.87 | 90.06 | 6.74 | 77.43 | 90.66 | 14.98 | 69.13 |
| FMP | 91.67 | **1.67** | **91.71** | 91.85 | **6.44** | **74.43** | 91.77 | **1.90** | **90.52** | 91.92 | **2.89** | **88.59** | 93.42 | **1.38** | **88.98** |

Table 7: Performance comparison (%) of other SOTA backdoor defense methods on CIFAR10 under PreActResNet18, under different attack strategies with a poison rate of 10% and retraining data ratio of 100%. We set the $\epsilon$ to 1/255, and the $p$ is set to 64.

# B    ABLATION STUDY ILLUSTRATION

In this section, we illustrate the figure illustration for the tables illustrated in Tab.3 to Tab.5.

**Effect of Poison Data Rate** The poison rate, referring to the proportion of poisoned samples in the training dataset, plays a crucial role in the results of the backdoor trigger injection. We conducted experiments with different poison rates (from 0.1% to 10%) to explore their impact on FMP's effectiveness. The results, shown in Fig.1, indicate that FMP demonstrates consistent performance across different poison rates and effectively mitigates backdoor attacks. For example, considering the BadNets attack, the ASR changes slightly within 0.99% to 1.78% as the poisoning rate increases from 0.1% to 10%. This trend is also observed for other attack strategies. Although a higher poison rate can be expected to lead to a higher ASR, our experimental results show that this is not true. When the poisoning rate is very low, it becomes more challenging for defense strategies to detect the backdoor trigger from the model due to its subtle influence. As the poison rate increases, the backdoor trigger has a more noticeable impact on the model, which can be detected and mitigated more easily by the defense strategy. Our experimental results emphasize the importance of designing defense strategies capable of detecting and mitigating backdoor attacks, even when dealing with subtle influences caused by low poison rates.

**Effectiveness under Different Percentages of Clean Data** We are also interested in studying the correlation between the performance of FMP and the amount of available training data, which will be used to repair the model to mitigate backdoor triggers. We compare four different retraining data ratios:5%, 10%, 15%, 20%, and 100%, and the results of our FMP are demonstrated in Fig.2. We observe that the performance of our defense strategy improves as the amount of clean training data increases. For example, when the retraining ratio increases from 5% to 100%, the ASR for BadNets decreases from 1.77% to 0.9%, while the model accuracy (Acc) improves from 86.57% to 92.02% and the Robust Accuracy (RA) increases from 86.08% to 91.04%. Similar trends can be observed for other attack strategies such as Blended, Low Frequency, SSBA, and WaNet. This indicates that our defense strategy becomes more effective in mitigating backdoor attacks as more clean data are available to retrain the model. However, it should be noted that even with a small amount of clean data (e.g., 5%), our defense strategy still exhibits competitively good performance in mitigating backdoor attacks. For example, with a 5% retraining ratio, the ASR for WaNet is 1.54%, while the Acc and RA are 89.31% and 88.07%, respectively.

**Effectiveness under Different $\epsilon$ and $p$** We further investigate the effectiveness of FMP under different $\epsilon$ and $p$, as listed in Fig.3 and Fig.4. We can first observe that with different $\epsilon$, the effectiveness of FMP is consistently satisfactory. Upon increasing the $\epsilon$, the Acc exhibits marginal decline, underscoring FMP's resilience across varying $\epsilon$ values. Subsequently, when varying the parameter $p$ for backdoor feature pruning, a notable decrease is observed in both accuracy (Acc) and robust accuracy (RA). This reduction can be attributed to the model's failure to successfully finetune after 50% of the information is pruned along with 50% of the feature maps, hampering its performance. FMP can successfully execute to mitigate the backdoor from the model with a $p$ larger than 4.

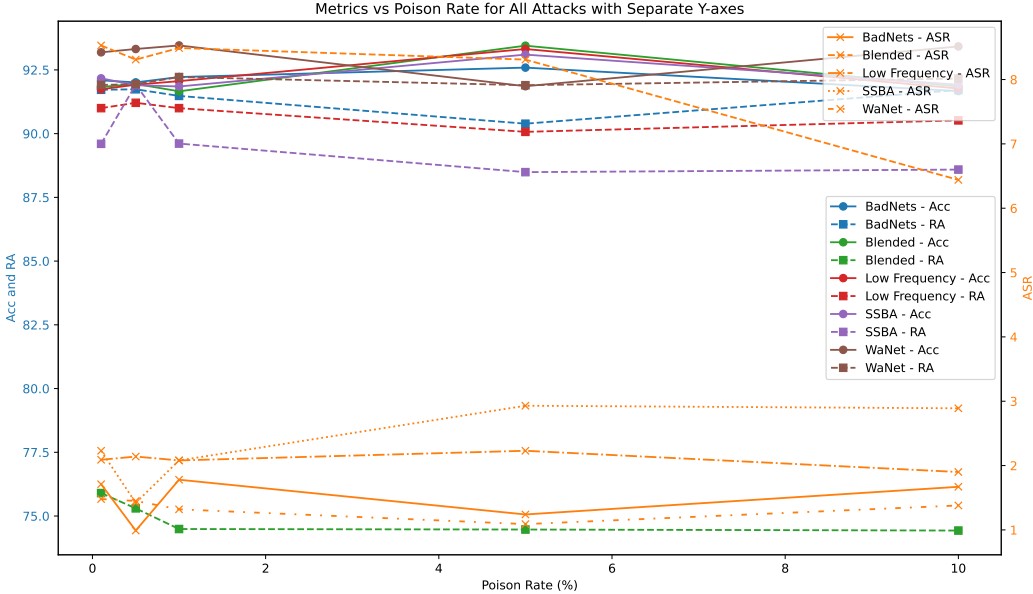

Figure 1: FMP's effectiveness under different poison rates.

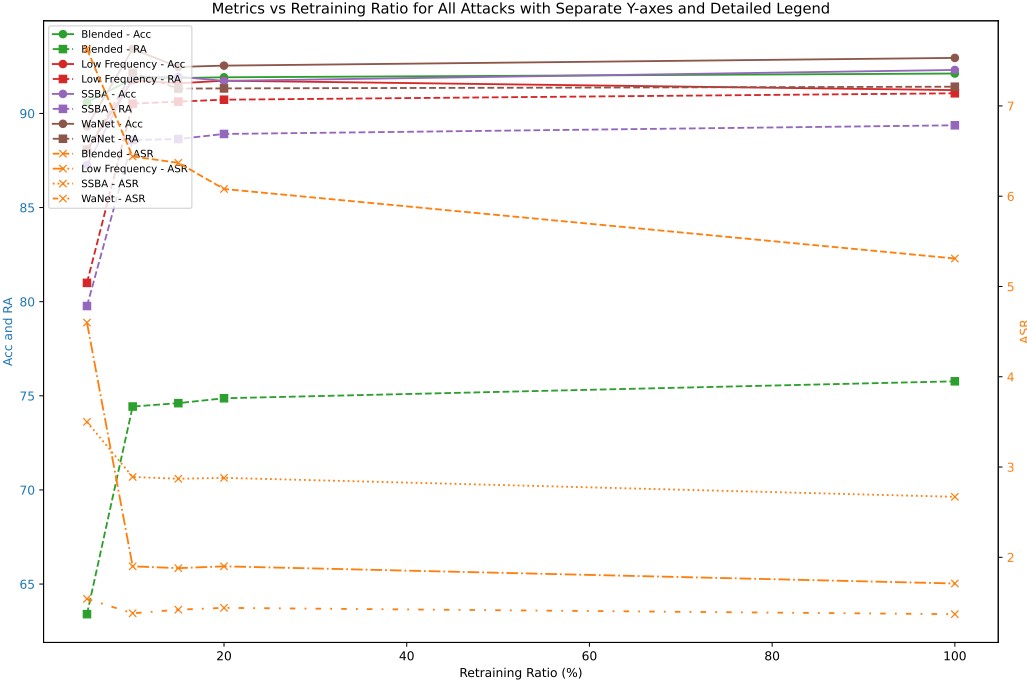

Figure 2: FMP's effectiveness under different retraining data ratios.

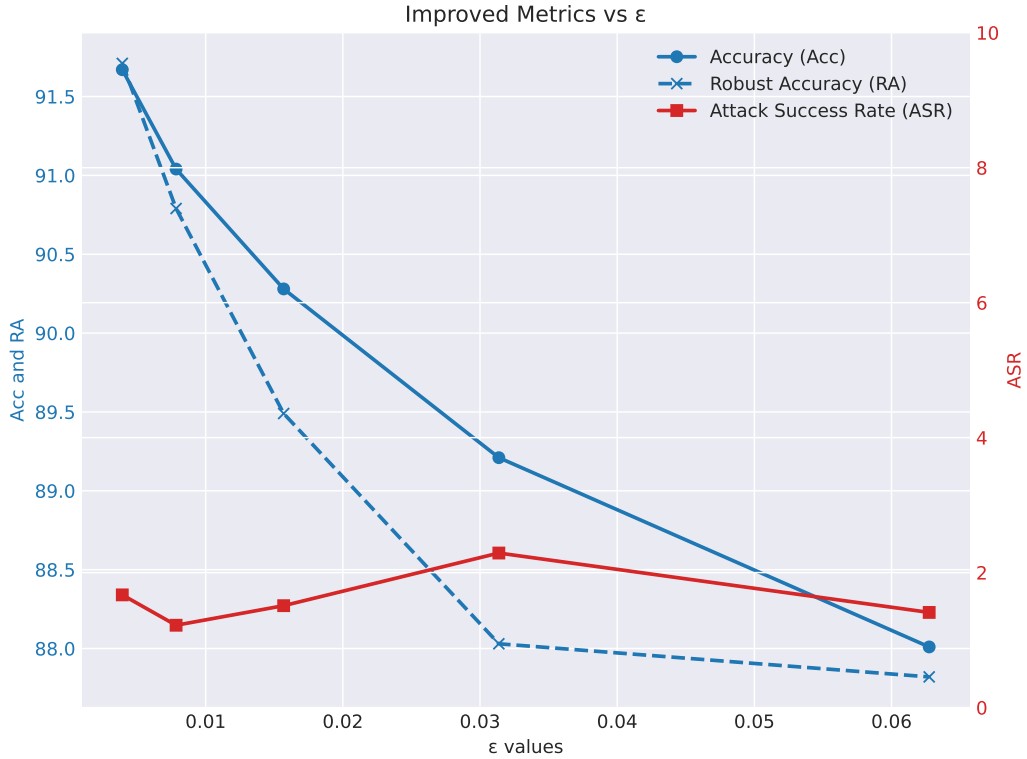

Figure 3: FMP's effectiveness under different $\epsilon$ in CIFAR10 dataset under BadNets attack.

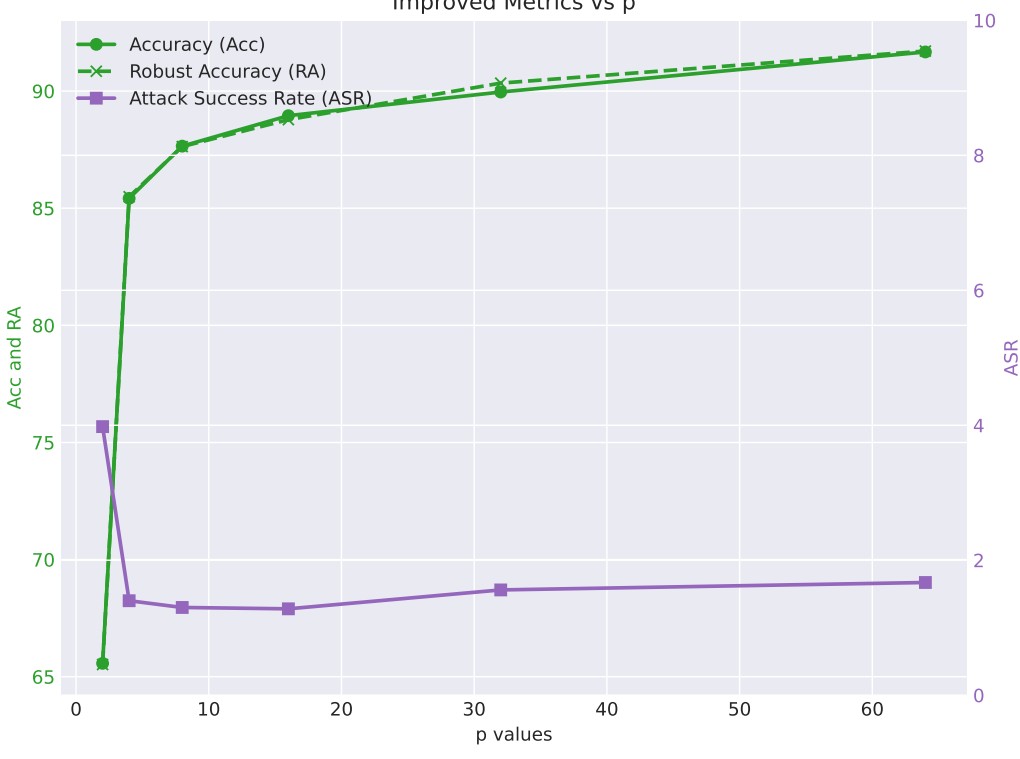

Figure 4: FMP's FMP's effectiveness under different $p$ in CIFAR10 dataset under BadNets attack.

