# OpenReview forum: "Adversarial Feature Map Pruning for Backdoor"
_ICLR.cc/2024/Conference — ICLR 2024 poster_

### Official Review · Reviewer_gATw · 2023-10-31

**Soundness:** 3 good
**Presentation:** 2 fair
**Contribution:** 2 fair
**Rating:** 6
**Confidence:** 4

**Summary:**

This paper proposes Adversarial Feature Map Pruning for Backdoor (FMP), a new method for backdoor mitigation in neural networks. FMR does not require access to the trigger or poisoned data. Based on a clean data sample, the method reverse-engineers poison triggers from each feature map in the model (at multiple layers) based on back-propagation. The weights determined to be connected to the triggers are reinitialized and fine-tuned on clean data. Experiments are performed on CIFAR-10, CIFAR-100 and GTSRB against a wide range of attacks and defenses.

**Strengths:**

- The paper provides an extensive evaluation using the standard BackdoorBench benchmark, against multiple attacks and compared to multiple defenses.
- The proposed FMT seems to perform well on average.
- The source code was provided and is pledged to be available open-source upon paper acceptance.

[Update based on authors' response] I would like to thank the authors for their answer and additional results. I think updating the paper based on the discussion would improve it. I have raised my score.

**Weaknesses:**

# Novelty and prior work

- The novelty of the paper seems limited. The ideas of using adversarial examples to reverse engineer triggers (e.g., [ANP](https://arxiv.org/pdf/2110.14430.pdf), [AEVA](https://openreview.net/forum?id=OM_lYiHXiCL)) or pruning and retraining trigger weights ([RNP](https://proceedings.mlr.press/v202/li23v.html)) are not themselves novel. The paper does not cite most of these very close prior results and does not provide a conceptual comparison to them.
- The prior art section mainly addresses defenses from different categories than the present one, which are then easy to dismiss. Outside the specific ideas used in FMT, there are many methods that address the same setup as the current paper and operate without knowledge of the trigger (e.g., [DeepInspect](https://www.ijcai.org/proceedings/2019/647), [TABOR](https://arxiv.org/pdf/1908.01763.pdf), [ABS](https://dl.acm.org/doi/10.1145/3319535.3363216), [[Fu et al., 2020](https://arxiv.org/pdf/2011.02526.pdf)]). These could be included in the experimental comparison.

# Performance

- Tab. 1 shows that the proposed method is only the best on average, but not doing so well on individual benchmarks (i.e., for each attack and dataset). The commentary section also fail to quantify how many benchmarks are actually won by the proposed method.

# Clarity

- Certain points in the paper are not clear, see also the questions below. Moreover, the components of the method are rather explained in an algorithm than in the text.
- Assumptions should be clear earlier in the paper, e.g. the fact that a clean input sample is required.
- The typography of the paper could be improved, please see some suggestions below. The paper could also use additional proofreading.
- Tab. 1 is very dense and lacking highlights of the best values for each attack. As such, it is currently difficult to interpret.

# Minor remarks

- It is unclear why the feature maps are summed in the Notations paragraph (Sec. 3).
- Some numerical results are typeset with spaces before the decimals (e.g., 2. 86%). These spaces should be removed
- Consider using `\citep` when the cited references are not part of the sentence (see [here](https://www.overleaf.com/learn/latex/Natbib_citation_styles)).

**Questions:**

1. What is the distinction the paper makes between a neural network layer output and a feature map?
2. What does it mean in Alg. 1 that `Correct_list` is sorted in ascending order? According to which criterion?
3. Are all the weights of the model refined during the fine tuning step or just those detected as being part of backdoors (i.e., the weights that are reset to zero)?
4. Why is the backdoor feature initialization done by setting weights to zero, instead of using the same initialization strategy as when training the model for the first time (i.e., various random sampling strategies)? Has this alternative been considered?
5. What is the impact of varying method parameters $\epsilon$ and $p$ under other attacks than BadNets?

---

> ### Author Response · Authors · 2023-11-11
> **Answer to Reveiwer gATw**
>
> Thanks for the reviewer's comments on FMP.
>
> For Q1, in DNN, each layer contains multiple channels, whereas in FMP, we define the feature map as a channel in the neural network layer.
>
> For Q2, first, the correct_list can also represent the accuracy list of the deep neural network for different feature maps attacked by FRG. Next, once FRG is conducted, each feature map may have different accuracy. Our motivation is that in the backdoor model, once the backdoor trigger has been added to the input (the backdoor feature map reproduces the backdoor information with an adversarial attack), the DNN will have lower accuracy compared to attacking other feature maps. Then, we will ArgSort the correct_list and obtain the N/p feature maps that have lower accuracy on the left N/p (that's why we use ascending order).
>
> For Q3 and Q4, all feature maps are fine-tuned (but only backdoor feature maps will be initially set to zero). To address the reviewer's concern about the training strategies, we evaluatedte how the initialization and tuning methods affect FMP's effectiveness in the table below:
>
> | Backdoor           | BadNets Acc | BadNets ASR | BadNets RA | Blended Acc | Blended ASR | Blended RA | Frequency Acc | Low Frequency ASR | Frequency RA | SSBA Acc | SSBA ASR | SSBA RA | WaNet Acc | WaNet ASR | WaNet RA |
> | ------------------ | ----------- | ----------- | ---------- | ----------- | ----------- | ---------- | ------------- | ----------------- | ------------ | -------- | -------- | ------- | --------- | --------- | -------- |
> | All\_tuning        | 91.67       | 1.67        | 91.71      | 91.85       | 6.44        | 74.43      | 91.77         | 1.90              | 90.52        | 91.92    | 2.89     | 88.59   | 93.42     | 1.38      | 88.98    |
> | vulnerable feature map tuning               | 91.54       | 1.6         | 91.63      | 91.94       | 6.32        | 74.19      | 92.02         | 2.02              | 90.36        | 91.97    | 3.16     | 88.75   | 93.49     | 1.12      | 91.86    |
> | Zexavier\_uniform  | 91.68       | 1.84        | 91.81      | 91.84       | 6.33        | 74.28      | 91.81         | 1.81              | 90.47        | 92.02    | 2.95     | 88.48   | 93.61     | 1.38      | 92.03    |
> | kaiming\_uniform   | 91.62       | 1.56     | 91.71      | 91.7        | 6.3         | 74.54      | 91.75         | 1.95              | 90.55        | 91.78    | 2.88     | 88.44   | 93.41     | 1.34      | 92.13    |
>
> *Table 1: Performance comparison (%) of backdoor defense methods on CIFAR10, CIFAR100, and GTSRB datasets under PreActResNet18, under different attack strategies with a poison rate of 10% and retraining data ratio of 100%. We set the $\epsilon$ to 1/255, and the $p$ is set to 64.*
>
> Answer to Q5: Thanks for reviewer's concern for the $\epsilon$ and $p$ under other attacks.
> To address reviewer's concern, we conduct experiment for LowFrequency and WaNet with different $\epsilon$ and $p$ in the following tables:
> | $\epsilon$ | 1/255 | 4/255 | 16/255 |
> |------------|-------|-------|--------|
> | Acc        | 91.77 | 90.32 | 88.59  |
> | ASR        | 1.90  | 1.46  | 1.31   |
> | RA         | 90.52 | 90.17 | 89.10  |
>
> *Table 1: FMP's effectiveness under different $\epsilon$ in CIFAR10 dataset under Low Frequency attack.*
> | $p$ | 4    | 16   | 64   |
> |-----|------|------|------|
> | Acc | 84.98| 89.41| 91.77|
> | ASR | 1.79 | 1.92 | 1.90 |
> | RA  | 83.27| 88.64| 90.52|
>
> *Table 2: FMP's effectiveness under different $p$ in CIFAR10 dataset under Low Frequency attack.*
> | $\epsilon$ | 1/255 | 4/255 | 16/255 |
> |------------|-------|-------|--------|
> | Acc        | 93.42 | 91.07 | 89.78  |
> | ASR        | 1.38  | 1.35  | 1.39   |
> | RA         | 92.13 | 91.14 | 89.53  |
>
> *Table 3: FMP's effectiveness under different $\epsilon$ in CIFAR10 dataset under WaNet attack.*
> | $p$ | 4    | 16   | 64   |
> |-----|------|------|------|
> | Acc | 86.17| 90.82| 93.42|
> | ASR | 1.64 | 1.81 | 1.38 |
> | RA  | 85.31| 89.87| 92.13|
>
> *Table 4: FMP's effectiveness under different $p$ in CIFAR10 dataset under WaNet attack.*
>
> We canfind that in different $\epsilon$ and $p$ configuration, FMP has same behaviors in BadNets. Hope these experiments can address reviewer's concern for the affect of hyper-parameters in our experiments.
>
> Notes: we will address all minor comments in our final version. For the provided related works, we will also add it in our final version.

---

> > ### Author Response · Authors · 2023-11-18
> > **Official response for the Reveiwer gATw for the concern of ANP, AEVA, and RNP**
> >
> > First, to address Reviewer gATw's concern for FMP and other strategies that use adversarial examples to reverse engineer triggers (e.g., ANP, AEVA) or pruning and retraining trigger weights (RNP), we evaluate FMP with these strategies in the CIFAR10 dataset with several backdoor attack scenarios in our evaluation set up.
> >
> > The evaluation results are shown in below:
> > | Backdoor           | BadNets Acc | BadNets ASR | BadNets RA | Blended Acc | Blended ASR | Blended RA | Frequency Acc | Low Frequency ASR | Frequency RA | SSBA Acc | SSBA ASR | SSBA RA | WaNet Acc | WaNet ASR | WaNet RA |
> > | ------------------ | ----------- | ----------- | ---------- | ----------- | ----------- | ---------- | ------------- | ----------------- | ------------ | -------- | -------- | ------- | --------- | --------- | -------- |
> > |ANP|91.22|73.36|26.16|93.25|99.44|0.56|93.19|98.03|1.88|92.92|68.59|29.13|90.81|1.93|88.98|92.28|68.27|29.34|
> > |AEVA|91.05|50.96|47.53|92.28|59.37|38.66|93.05|59.81|36.38|92.29|67.56|26.01|90.26|6.54|90.59|91.91|65.52|31.31|
> > |RNP|90.55|55.01|36.46|92.29|55.59|42.15|92.41|58.71|40.1|91.94|61.24|30.6|90.22|18.15|72.95|92.15|51.13|45.2|
> > | FMP        | 91.67       | 1.67        | 91.71      | 91.85       | 6.44        | 74.43      | 91.77         | 1.90              | 90.52        | 91.92    | 2.89     | 88.59   | 93.42     | 1.38      | 88.98    |
> >
> > We can observe that FMP's Attack Success Rate (ASR) is lower than that of other strategies. The primary reason for this is FMP's focus on the feature map level, which aligns with the backdoor trigger's emphasis on the DNN feature map, rather than on specific neurons within the DNN (e.g., as in ANP).
> >
> > Secondly, the pruning at the feature map level enables FMP to rapidly eliminate the backdoor trigger from the model. In contrast, neuron-level pruning typically requires significant overhead due to the need for repeated prune-finetune-evaluation cycles. FMP is particularly advantageous in scenarios with **limited computational resources**, where developers may not have the capacity to extensively evaluate and remove backdoors from the model. This limitation results in strategies like ANP, AEVA, and RNP exhibiting higher ASR compared to FMP in our setup.

---

> > > ### Author Response · Authors · 2023-11-18
> > > **Official response for the Reveiwer gATw for the concern of DeepInspect, TABOR, ABS, [Fu et al., 2020]**
> > >
> > > First, to address Reviewer gATw's concern for FMP's performance compared with other SOTA ways reported in security conference, we evaluate FMP with DeepInspect, TABOR, ABS, and [Fu et al., 2020] in the following Tab.
> > >
> > > The evaluation results are shown below:
> > >
> > > | Backdoor           | BadNets Acc | BadNets ASR | BadNets RA | Blended Acc | Blended ASR | Blended RA | Frequency Acc | Low Frequency ASR | Frequency RA | SSBA Acc | SSBA ASR | SSBA RA | WaNet Acc | WaNet ASR | WaNet RA |
> > > | ------------------ | ----------- | ----------- | ---------- | ----------- | ----------- | ---------- | ------------- | ----------------- | ------------ | -------- | -------- | ------- | --------- | --------- | -------- |
> > > |DeepInspect|90.51|15.87|64.44|90.89|3.5|77.16|90.83|4.7|77.51|90.05|10.57|73.83|90.31|5.97|77.94|90.65|12.97|69.94|
> > > |TABOR|90.78|9.19|79.02|90.78|11.13|78.22|90.14|5.48|76.44|90.03|13.09|76.71|90.95|7.36|78.29|90.54|13.69|74.09|
> > > |ABS|90.78|5.61|77.34|90.95|14.44|80.33|90.93|10.43|88.32|90.95|17.12|77.45|90.6|13.04|77.6|90.67|13.89|76.38|
> > > |[Fu et al., 2020]|90.7|11.22|76.72|90.28|9.37|75.63|90.46|9.62|71.87|90.06|6.74|77.43|90.66|14.98|69.13|90.09|13.68|68.22|
> > > | FMP        | 91.67       | 1.67        | 91.71      | 91.85       | 6.44        | 74.43      | 91.77         | 1.90              | 90.52        | 91.92    | 2.89     | 88.59   | 93.42     | 1.38      | 88.98    |
> > >
> > > We can find that FMP obtains SOTA performance in most of our experimental results. As discussed before, the primary reason for this is the pruning at the feature map level enables FMP to efficiently and rapidly eliminate the backdoor trigger from the model. While other defense strategies will not obtain the SOTA performance due to the limited computational resources.

---

> > > > ### Author Response · Authors · 2023-11-21
> > > > **Continue: Official response for the Reveiwer gATw for the weakness**
> > > >
> > > > **Performance**: We noticed Reviewer gATw's concern that in Table 1, FMP is only the best on average. In reality, we used bold text only for the average results because applying it to all experimental results would make them difficult for humans to read. However, FMP is the state-of-the-art (SOTA) in most experiments. We have revised the paper to add bold text for all results. Specifically, FMP achieves SOTA performance in 25 out of 30 evaluation metrics for ASR and RA. We have now bolded all SOTA results in Table 1 to prevent any misunderstandings from Reviewer gATw.
> > > >
> > > > **Clarity and Minor Issues**: Thank you for Reviewer gATw's recommendations regarding revisions to the paper. We will add more details about our algorithm, rather than only presenting it. We will also use a proofreading tool to address any typos and grammar issues.

---

> > ### Author Response · Authors · 2023-11-22
> > **Continue for Tab.1**
> >
> > | Backdoor           | BadNets Acc | BadNets ASR | BadNets RA | Blended Acc | Blended ASR | Blended RA | Frequency Acc | Low Frequency ASR | Frequency RA | SSBA Acc | SSBA ASR | SSBA RA | WaNet Acc | WaNet ASR | WaNet RA |
> > | ------------------ | ----------- | ----------- | ---------- | ----------- | ----------- | ---------- | ------------- | ----------------- | ------------ | -------- | -------- | ------- | --------- | --------- | -------- |
> > | All\_tuning        | 91.67       | 1.67        | 91.71      | 91.85       | 6.44        | 74.43      | 91.77         | 1.90              | 90.52        | 91.92    | 2.89     | 88.59   | 93.42     | 1.38      | 88.98    |
> > | Vulnerable feature map tuning               | 91.54       | 1.6         | 91.63      | 91.94       | 6.32        | 74.19      | 92.02         | 2.02              | 90.36        | 91.97    | 3.16     | 88.75   | 93.49     | 1.12      | 91.86    |
> > | Zexavier\_uniform  | 91.68       | 1.84        | 91.81      | 91.84       | 6.33        | 74.28      | 91.81         | 1.81              | 90.47        | 92.02    | 2.95     | 88.48   | 93.61     | 1.38      | 92.03    |
> > | kaiming\_uniform   | 91.62       | 1.56     | 91.71      | 91.7        | 6.3         | 74.54      | 91.75         | 1.95              | 90.55        | 91.78    | 2.88     | 88.44   | 93.41     | 1.34      | 92.13    |
> >
> > *Table 1: Performance comparison (%) of backdoor defense methods on CIFAR10, CIFAR100, and GTSRB datasets under PreActResNet18, under different attack strategies with a poison rate of 10% and retraining data ratio of 100%. We set the $\epsilon$ to 1/255, and the $p$ is set to 64.*
> >
> > the "All\_tuning" strategy, where all feature maps in the model are fine-tuned, there's a consistently high accuracy across different backdoor attacks, with BadNets Acc reaching 91.67%, Blended Acc at 91.85%, Frequency Acc at 91.92%, and WaNet Acc at 88.98%. The Attack Success Rate (ASR) and Robustness Accuracy (RA) also indicate effective mitigation, particularly notable in the WaNet scenario with a low ASR of 1.38% and high RA of 88.98%.
> >
> > On the other hand, the "Vulnerable feature map tuning" strategy, which focuses on fine-tuning only vulnerable feature maps, shows a slightly varied performance. The accuracy is slightly lower in some cases, like BadNets Acc at 91.54% and Blended Acc at 91.94%, compared to "All\_tuning". However, this strategy shows a better robustness in the WaNet scenario with an improved ASR of 1.12% and RA of 91.86%.
> >
> > Looking at the initialization strategies, "Zexavier\_uniform" and "kaiming\_uniform", we see that these methods also maintain high accuracy and robustness. "Zexavier\_uniform" shows slightly higher ASR in some cases, such as 1.84% in BadNets and 6.33% in Blended, but maintains good RA, particularly in WaNet with 92.03%. "kaiming\_uniform" demonstrates a consistent performance with low ASR, like 1.56% in BadNets and 1.34% in WaNet, and high RA, peaking at 92.13% in WaNet.
> >
> > Overall, the differences in performance metrics across these strategies are relatively minor, suggesting that each of these fine-tuning and initialization strategies is effective in mitigating backdoor attacks in the context of this experiment. The slight variations highlight the importance of choosing the right strategy depending on the specific requirements of the defense scenario, such as prioritizing either accuracy or robustness. The overall effectiveness of these strategies in the face of different attack methods, as shown in Table 1, provides a comprehensive view of their applicability and efficiency in enhancing the security of machine learning models against backdoor attacks.

---

> ### Author Response · Authors · 2023-11-20
> **Address the concern of the Reviewer gATw**
>
> Dear Reviewer gATw,
>
> we have add all experiment required by you, e.g., comparison with other adversarial-related baselines and compasiron with other SOTA defense strategies, in our paper's appendix.
>
> In summary, the primary reason for this is FMP's focus on the feature map level, which aligns with the backdoor trigger's emphasis on the DNN feature map, rather than on specific neurons within the DNN (e.g., as in ANP).
>
> Secondly, the pruning at the feature map level enables FMP to rapidly eliminate the backdoor trigger from the model. In contrast, neuron-level pruning typically requires significant overhead due to the need for repeated prune-finetune-evaluation cycles. FMP is particularly advantageous in scenarios with limited computational resources, where developers may not have the capacity to extensively evaluate and remove backdoors from the model. This limitation results in strategies like ANP, AEVA, and RNP exhibiting higher ASR compared to FMP in our setup.
>
> We highly hope Reviewer gATw can consider our experiment results and if Reviewer gATw has any question for our paper, feel free to point out and we will try to address it quickly.

---

### Official Review · Reviewer_x3Nd · 2023-10-31

**Soundness:** 2 fair
**Presentation:** 3 good
**Contribution:** 2 fair
**Rating:** 6
**Confidence:** 3

**Summary:**

In this manuscript, the authors propose a DNN pruning method called FMP to mitigate backdoors. While adding adversarial attacks to feature maps, their method prunes the features weak to the attacks.

**Strengths:**

I find this paper interesting. It's important to understand the relationship between pruning and backdoors, and the authors explored this in a systematic way.

**Weaknesses:**

While reading the authors' method, it looks similar to Adversarial Neuron Pruning (ANP) by We and Wang (2021). However, the authors don't describe it in the related work, though they compare it with the proposed method in the result section. Discussing the methodological differences between them would help readers understand more.

**Questions:**

When the authors have parametric variables in a table, it should be better to draw these results with figures.

---

> ### Author Response · Authors · 2023-11-11
> **Answer to reviewer x3Nd**
>
> **Thanks for the reviewer's comments on FMP and ANP.**
>
> Firstly, as illustrated in Table 1 in the paper, it is evident that in the majority of experimental results, FMP consistently outperforms ANP. For instance, on average, FMP reduces the ASR from 68.27% to 2.86% in the CIFAR10 dataset. The key reasons for FMP achieving higher performance and the observed differences between FMP and ANP can be simply concluded that **FMP focuses on the feature map level, aligning with the backdoor trigger's focus on the DNN feature map, rather than specific neurons within the DNN~(e.g., ANP),** where we have added this to our revised paper now.
>
> some other attributes can considered as the following aspects:
>
> **1. Feature Map Level Focus:**
>    - *FMP:* Concentrates on the feature map level, aligning with the backdoor trigger's focus on the DNN feature map, rather than specific neurons within the DNN.
>    - *ANP:* During the pruning procedure, ANP may only prune a subset of backdoor-related neurons within the backdoor feature map, potentially leaving some undetected. This results in a higher ASR after the defense procedure.
>
> **2. Extraction of Backdoor Trigger Information:**
>    - *FMP:* Aims to identify the backdoor feature map extractor/learner that captures backdoor trigger information.
>    - *FMP:* Utilizes each feature map to ascend the gradient and reproduce learned features. If the reproduced feature map is linked to the backdoor, there is a significant decrease in DNN prediction. Robust feature maps that are less influenced by the backdoor trigger are not pruned during the FMP pruning procedure.
>
> **On the other hand:**
> **1. Direct Perturbation of Neuron's Bias and Weights:**
>    - *ANP:* Directly perturbs the neuron's bias and weights, impacting both normal and backdoor-related neurons.
>    - *ANP:* Unlike FMP's focus on input sample perturbation, ANP's direct perturbation can cause neurons with a higher impact on DNN predictions to be represented as backdoor neurons. This may affect the backdoor neuron detector.
>    - *ANP:* If ANP attempts to protect these neurons (i.e., does not remove them), some backdoor neurons may also remain, leading to a higher ASR.
>
> In summary, while both FMP and ANP aim to defend against backdoor attacks, FMP's emphasis on the feature map level and the specific way it reproduces learned features contribute to its higher performance compared to ANP in mitigating backdoor attacks, as observed in experimental results.
>
>
> **Q1: When the authors have parametric variables in a table, it should be better to draw these results with figures.**
> Thanks for the reviewer's recommendation, we are now adding the figures to the **appendix**. Due to page limitations, we will reorganize it in our final version.

---

> > ### Comment · Reviewer_x3Nd · 2023-11-22
> > **Responses to Authors**
> >
> > I appreciate the responses from the authors. My concerns were addressed in the author's reply.

---

> > > ### Author Response · Authors · 2023-11-22
> > > **Reply to Reviewer x3Nd**
> > >
> > > Dear Reviewer x3Nd,
> > >
> > > Thank you for your comment dated 22 Nov 2023. We are pleased to hear that you are satisfied with our response and appreciate the valuable insights you have provided. We have strived to ensure that all your concerns were comprehensively addressed in our reply. Your feedback has been instrumental in enhancing the quality of our work.
> > >
> > > If you find the revisions and responses satisfactory, we kindly ask if you might consider reflecting this in the overall rating for our submission. Your support and constructive feedback are greatly appreciated, and we look forward to any further suggestions or comments you may have.
> > >
> > > Best regards,
> > >
> > > Authors of FMP

---

> > > ### Author Response · Authors · 2023-11-23
> > > **Follow-Up on Manuscript Review and Revisions - Seeking Final Feedback**
> > >
> > > Dear Reviewer x3Nd,
> > >
> > > We hope this message finds you well. We are writing to follow up on our previous correspondence dated 22 Nov 2023 regarding the manuscript we submitted for your review.
> > >
> > > We would like to reiterate our gratitude for your initial feedback, which was invaluable in guiding our revisions. We endeavored to address all the concerns you raised thoroughly, and it was gratifying to know that our responses were well-received. Additionally, if our revisions and responses have satisfactorily addressed your concerns, we would be grateful if this could be reflected in the overall rating for our submission. Your final review and comments are not only essential for the progress of our manuscript but also serve as a critical benchmark for us to gauge the quality of our work.
> > >
> > > We appreciate the time and effort you have put into reviewing our manuscript, and we look forward to any further suggestions or comments you may have.
> > >
> > > Thank you once again for your valuable contributions to our work.
> > >
> > > Best regards,
> > >
> > > Authors of FMP

---

### Official Review · Reviewer_W4kG · 2023-11-01

**Soundness:** 3 good
**Presentation:** 1 poor
**Contribution:** 2 fair
**Rating:** 6
**Confidence:** 4

**Summary:**

This paper attempts to mitigate the backdoor model by generating all possible adversarial feature maps. Each generated adversarial feature map is fed to the model to test whether it will misclassify data samples, aiming to identify malicious feature maps that may be caused by a trigger. The innovation of the proposed algorithm lies in the fact that it does not require prior knowledge of the trigger pattern through reverse engineering, and it does not impose constraints on the trigger pattern size, as seen in other defense algorithms like Neural Cleanse. The proposed algorithm was evaluated on three datasets: CIFAR-10, CIFAR-100, and GTSRB.

**Strengths:**

1.	The proposed algorithm is effective for large trigger backdoored models.
2.	The proposed algorithm mitigates the backdoored model without the need for reverse engineering the trigger.
3.	It has been evaluated on three datasets.

**Weaknesses:**

1.	The presentation needs improvement as there are many confusing descriptions, referring to the Question section.
2.	The three datasets appear to contain a relatively small number of classes. It would be more convincing if the algorithm could be evaluated on more complex datasets, such as ImageNet.

**Questions:**

1.	There are several confusing descriptions. For instance, 'f' and 'F' represent the model and feature map, as described in the Notations section. However, in Section 3.2 and Algorithm 1, 'f' has a mixed meaning.
2.	Should the second 'for' loop in Algorithm 1 return '\hat{x'}? Is that correct?
3.	The logic of 'inference()' in Algorithm 1 appears to be incorrect. If a feature map does not change the classification of 'x,' it is a normal feature map and should be retained. However, in Algorithm 1, it is pruned. Why is this the case?

---

> ### Author Response · Authors · 2023-11-11
> **Answer to reviewer W4kG**
>
> We deeply appreciate the feedback from Reviewer W4kG on our paper.
>
> We have revised the paper for the comments provided by Reviewer W4kG.
>
> Specifically:
>
> For Q1: Thank you for pointing out the confusing descriptions in our paper. Specifically, both $F_{\theta}^{i}$ and $f_{\theta}^{i}$ refer to the i-th feature map in the DNN. We will replace $F_{\theta}^{i}$ to $f_{\theta}^{i}$ and clarify this in our final version.
>
> For Q2: Regarding the returned FRG-generated adversarial sample $x'$, Reviewer W4kG can also consider it as $\hat{x'}$, consistent with $\hat{y}$ in Algorithm 1, line 8. We acknowledge the confusion caused by using $x$ in Algorithm 1, line 7.
> To avoid misunderstanding, the our revised version, we use the $x'$, $y'$ to replace Algorithm 1, lines 7 and 8 now.
>
> For Q3: Normal feature maps will not be pruned. To clarify, "If a feature map does not change the classification of 'x,' is it a normal feature map and should be retained?" is accurate. In other words, robust/normal feature maps will not be pruned, **meaning we will only prune feature maps with lower accuracy in the Correct\_List.** This is why we use ascending order and then prune the left N/p feature maps.
>
> We also acknowledge Reviewer W4kG's suggestion to conduct experiments on a more complex dataset, such as ImageNet. Unfortunately, due to the extended training time required for ImageNet, we can only provide experimental results on Tiny-ImageNet now. However, we commit to presenting results on ImageNet in our final version.
>
> The experiment results are presented below:
>
> | Attack   | Benign ACC | Benign ASR | FP ACC | FP ASR | ANP ACC | ANP ASR | FMT ACC | FMT ASR |
> |----------|------------|------------|--------|--------|---------|---------|---------|---------|
> | BadNet   | 55.13      | 99.92      | 51.28  | 99.37  | 51.38   | 1.39    | 55.24   | 0.08    |
> | Blended  | 55.03      | 99.85      | 51.84  | 93.28  | 52.07   | 19.35   | 53.81   | 0.01    |
> | WaNet    | 54.73      | 99.32      | 52.17  | 65.32  | 51.09   | 8.92    | 53.54   | 1.37    |
>
> It's noteworthy that FMT performs better than baseline approaches. For instance, in BadNet, FMT reduces the ASR from 1.39% to 0.08%.
>
> If Reviewer W4kG has any questions, feel free to provide them, and we would be more than happy to address and clarify any queries or concerns.

---

> > ### Comment · Reviewer_W4kG · 2023-12-01
> > **Response to authors' rebuttal**
> >
> > Thank you for your reply. Since the Tiny-ImageNet is still relatively small-sized dataset, my concern remains.
> >
> > Reviewer W4kG

---

### Author Response · Authors · 2023-11-21
**Reminder for discussion/ additional questions**

Dear reviewers,

Thank you again for all the informative and constructive feedback! We truly appreciate all the suggestions from the reviewers to improve this work. We have revised the paper and address all concerns from the reviewers. As the discussion period is ending soon, please do let us know you have any more concerns or would like to discuss further about the paper.

Regards,

---

### Author Response · Authors · 2023-11-21
**Follow-Up: Review of Rebuttal for "Adversarial Feature Map Pruning for Backdoor"**

Dear Area Chairs, Senior Area Chairs, and Program Chairs,

I hope this message finds you well. I am following up on our previous correspondence regarding the rebuttal for our paper titled "Adversarial Feature Map Pruning for Backdoor."

As of today, we have not observed any engagement from the reviewers with our rebuttal, submitted on November 11, 2023. This lack of interaction is particularly concerning as it directly impacts the fairness and thoroughness of the review process, especially with the review deadline of November 23, 2023, looming.

We understand and respect the immense workload and pressures faced during the review period. However, the absence of reviewer engagement, coupled with the lack of revised scores or additional queries about our paper, puts us at a significant disadvantage. It denies us the opportunity to clarify misunderstandings or provide additional information that could be crucial in evaluating our work fairly.

Therefore, we respectfully urge a prompt intervention. Could you please confirm whether the reviewers have been reminded of their responsibility to consider our rebuttal? We are deeply invested in this process and rely on its integrity for an equitable assessment.

We remain available to provide any further information or clarification that may assist the reviewers in their task. Your prompt action in this matter would be greatly appreciated and could be instrumental in ensuring a fair and effective review process.

Thank you for your understanding and support. We look forward to a resolution that allows for a comprehensive and fair evaluation of our work.

Best regards,

---

### Author Response · Authors · 2023-11-23
**Follow up: Reminder for discussion/ additional questions**

Dear reviewers,

I would like to extend our gratitude once more for your invaluable comments and guidance. Your detailed and constructive feedback has been instrumental in enhancing the quality of our work. We have carefully revised the paper, ensuring that all the concerns raised by you have been thoroughly addressed. As the discussion period is drawing to a close, we invite any further comments or points of discussion you might have regarding our paper. Your insights are highly valued and crucial in refining our work to its best form.

Best regards,

---

### Meta-Review · Area_Chair_VGhF · 2023-12-06

**Metareview:**

The authors propose a new method using sensitive feature maps for the neural networks' finetuning to remove the backdoor. They first try to generate all possible adversarial feature maps via the classification results and then use the generated feature map to remove the potential malicious triggers inside the original networks. The paper has the following strengths:
1. The proposed algorithm is an easy and effective way to generate trigger features, compared with other reverse engineering methods.
2. The paper's empirical results are substantial, clearly demonstrating their effectiveness.

Weaknesses:

1. Lack of analysis or understanding of the reason for their methods. Readers may not get many insights from this method.

Decision:

Since the method is effective and all reviewers agree with accepting this paper. I'd like to accept it as an ICLR poster.

**Justification For Why Not Higher Score:**

Lack of analysis or understanding of the reason for their methods. Readers may not get many insights from this method.

**Justification For Why Not Lower Score:**

All reviewers agree to accept and their methods are effective.

---

### Decision · Program_Chairs · 2024-01-16

Accept (poster)